# Increased Glycine-*N*-methyltransferase expression disrupts light-dependent gene expression rhythms in the *Drosophila* eye

Seth D. Lammert[1], Makayla N. Marlin[1], Danny M. Kanj[1] and Vikki M. Weake[1,2,3,*]

## ABSTRACT

Glycine-*N*-methyltransferase (Gnmt) is highly upregulated in the aging *Drosophila melanogaster* eye. Gnmt regulates *S*-adenosylmethionine (SAM) and *S*-adenosylhomocysteine (SAH) availability by catalyzing the transfer of a methyl group from SAM to glycine, producing sarcosine and SAH. Gnmt is well studied in the liver; however, little is known about the consequences of increased Gnmt in neurons. We overexpressed Gnmt in the eye and profiled diurnal rhythmic gene expression and histone methylation in photoreceptor neurons. Here, we show that eye-specific Gnmt overexpression altered rhythmic gene expression without impacting photoreceptor viability. Gnmt overexpression decreased global repressive histone methylation but had no major effect on H3K4 methylation, suggesting that methylation reactions are selectively inhibited by Gnmt and that Gnmt disrupts rhythmic gene expression independently of H3K4 methylation. Gnmt overexpression did not alter rhythmicity of core clock genes and did not impact circadian behavior. These results suggest that Gnmt plays a role in the regulation of light-dependent rhythmic gene expression in photoreceptors that does not involve the molecular clock.

KEY WORDS: Gnmt, Circadian rhythms, One-carbon metabolism, Methylation potential, Photoreceptor

## INTRODUCTION

Methylation of proteins and nucleic acids regulate a variety of cellular processes including gene expression, RNA processing, purine synthesis and protein stability. Methyltransferases utilize *S*-adenosylmethionine (SAM) derived from one-carbon metabolism to methylate substrates, resulting in the byproduct *S*-adenosylhomocysteine (SAH). Because SAH is a potent inhibitor of SAM-dependent methyltransferases, the balance of SAM and SAH can impact methyltransferase activity (Hoffman et al., 1979). Glycine-*N*-methyltransferase (Gnmt), an enzyme that utilizes SAM for the methylation of glycine into sarcosine, plays a crucial role in regulating levels of SAM and SAH because it is much less sensitive to SAH inhibition compared with most other methyltransferases (Gnmt $K_i$=35-80 µM, tRNA methyltransferase

[1]Department of Biochemistry, Purdue University, West Lafayette, IN 47907, USA. [2]Purdue Center for Cancer Research, Purdue University, West Lafayette, IN 47907, USA. [3]Purdue Institute for Integrative Neuroscience, Purdue University, West Lafayette, IN 47907, USA.

*Author for correspondence (vweake@purdue.edu)

S.D.L., 0000-0002-8859-1495; M.N.M., 0009-0005-0246-1358; D.M.K., 0009-0007-9548-0711; V.M.W., 0000-0002-5933-9952

$K_i$=0.4 µM) (Kerr, 1972; Kerr and Heady, 1974). Thus, Gnmt influences the activity of other methyltransferases by outcompeting them for SAM at higher levels of SAH (Kerr, 1970, 1972). Gnmt is most abundant in the liver where its major function is to regulate systemic SAM levels (Luka, 2008). This is apparent in both mice and flies, as Gnmt-deficient individuals have drastically increased SAM and decreased SAH levels (Luka et al., 2006; Obata et al., 2014). Further, knockdown or overexpression of Gnmt in the *Drosophila* fat body, which plays an analogous role to the liver, increases or decreases whole-organism SAM, respectively, emphasizing that Gnmt expression level contributes to the availability of one-carbon units (Obata and Miura, 2015; Obata et al., 2014).

Independent of its catalytic activity, Gnmt also impacts one-carbon metabolism via the binding of 5-methyltetrahydrofolate (5-methylTHF), a substrate for methionine synthesis and the most abundant form of folate in the liver (Cook and Wagner, 1984). Binding of 5-methylTHF to Gnmt noncompetitively inhibits Gnmt activity and sequesters 5-methylTHF, preventing SAM consumption by Gnmt and the re-methylation of homocysteine into methionine (Wagner et al., 1985). Additionally, SAM allosterically inhibits 5-methylTHF synthesis, serving as a negative-feedback mechanism (Daubner and Matthews, 1982). Therefore, the activity of Gnmt and its regulation are crucial for balancing methyl units between the folate and one-carbon cycles for use in methylation reactions.

One-carbon metabolites exhibit diurnal rhythmic oscillations in the mouse liver, following roughly 24 h-long periods (Krishnaiah et al., 2017). When these ~24-h rhythms persist under constant environmental conditions, they are considered 'circadian'. This oscillation comes from precise timing of metabolism through a transcriptional regulatory network in response to feeding behaviors (Krishnaiah et al., 2017). Dietary intake is a zeitgeber ('time-giver'), an environmental stimulus that provides cues that entrain the timing of biological rhythms (Patke et al., 2020). The importance of dietary intake as a zeitgeber is apparent in flies, in which mistimed feeding disrupts behavioral circadian rhythms and rhythmic gene expression (Xu et al., 2011). Importantly, one-carbon metabolism itself plays a role in maintaining circadian rhythms. For example, dietary supplementation of SAM or decreased dietary folate can perturb behavioral circadian rhythms and rhythmic gene expression in mice (Challet et al., 2013; Fukumoto et al., 2022). Additionally, knockdown of individual genes in folate metabolism disrupts rhythmic gene expression in U2OS cells (Zhang et al., 2009). Circadian rhythms are primarily controlled by a core transcription-translation feedback loop (TTFL), often referred to as the molecular clock, that influences the timing and rhythmic expression of downstream genes (Patke et al., 2020). Based on the relationship between metabolism and gene expression, epigenetic mechanisms are postulated to connect one-carbon metabolism to circadian rhythms through factors such as histone methylation (González-Suárez and Aguilar-Arnal, 2024). For example, the one-carbon enzyme adenosylhomocysteinease (AHCY) is essential for rhythmic gene expression and it associates with circadian

clock transcriptional activators, promoting H3K4 methylation in mice (Greco et al., 2020; Stanhope and Weake, 2026).

Gnmt is highly upregulated during aging in flies, where it plays a protective role (Obata and Miura, 2015), promoting survival under stress conditions such as starvation (Obata et al., 2014). Interestingly, overexpression of Gnmt extends lifespan in flies but this only requires overexpression in the fat body (Obata and Miura, 2015; Tain et al., 2020). Age-dependent upregulation of Gnmt is not limited to the fat body, as Gnmt is one of the most highly upregulated proteins in aging fly eyes, including in photoreceptor neurons (Hall et al., 2017, 2021). Photoreceptor neurons, like the fat body, are highly metabolically active and require substantial amounts of energy but the two tissues differ in that the fat body utilizes energy for nutrient processing and synthesis of metabolites for use in other tissues, whereas the majority of energy used by photoreceptors is consumed for phototransduction (Laughlin et al., 1998; Okawa et al., 2008). Alterations to one-carbon metabolism in the liver, or fat body, therefore, have consequences for other tissues and organs, as seen in *Drosophila* where manipulation of Gnmt expression in the fat body can impair tissue repair in *Drosophila* imaginal discs (Kashio et al., 2016). Gnmt has been suggested to play a neuroprotective role because its overexpression in glial cells of mammalian neuron-glial cell culture improves survival and neurogenesis (Tsai et al., 2010) and $Gnmt^{-/-}$ mice have impaired neural progenitor cell proliferation and memory (Carrasco et al., 2014). This raises the question of what the biological consequences of increased Gnmt are in *Drosophila* neurons.

We previously found that global histone methylation decreased in aging *Drosophila* photoreceptor neurons, suggesting that methylation reactions may become less efficient with age (Jauregui-Lozano et al., 2023; McGovern et al., 2026). Further, profiling of the aging eye metabolome identified altered levels of metabolites involved in folate biosynthesis and SAM metabolism (Hall et al., 2021). Since increased Gnmt can alter one-carbon metabolism and potentially influence histone methylation, we hypothesized that overexpression of Gnmt in the eye decreases global histone methylation levels and disrupts rhythmic gene expression. In this study, we demonstrate that Gnmt expression in the eye does not impact age-dependent retinal degeneration or levels of active histone methylation. Rather, increased Gnmt disrupts light-dependent, rhythmic gene expression and decreases repressive histone methylation in the eye. However, our data show that overexpression of Gnmt in clock cells did not disrupt circadian locomotor behavior, suggesting that increased expression of Gnmt in photoreceptors impacts light-dependent, rhythmic gene expression but not the molecular clock.

## RESULTS
### Gnmt increase in heads of old *Drosophila* is accompanied by changes in methionine cycle metabolites
One-carbon metabolism supplies methyl units in the form of SAM for methylation reactions by methyltransferases (Fig. 1A). Gnmt catalyzes the transfer of a methyl group from SAM to glycine, producing sarcosine and SAH. We previously showed using RNA sequencing (RNA-seq) and proteomic analysis that Gnmt levels are increased at both transcript and protein level in aging *Drosophila* eyes (Hall et al., 2017, 2021). We confirmed this increase in Gnmt abundance by western blot analysis of head extracts from 10-day-old ('young') and 50-day-old ('old') flies, observing a 19-fold age-dependent increase in Gnmt levels (Fig. 1B,C). We then investigated whether increased Gnmt correlated with age-dependent changes in this tissue for metabolites associated with one-carbon metabolism, such as methionine, SAM and SAH. We examined levels of these metabolites by targeted metabolite analysis in head extracts from young and old

flies. While methionine, SAM and SAH levels all significantly increased at day (D) 50 relative to D10, glycine and sarcosine levels showed no significant difference (Fig. 1D). Similar age-dependent increases in SAM levels have been observed in whole bodies from old flies but were accompanied by decreases in methionine and sarcosine (Obata and Miura, 2015). Thus, we conclude that the increased Gnmt levels in old heads correlate with increased SAH but are accompanied by other changes in one-carbon metabolism, potentially indicating altered flux through this pathway in the aging head with changes distinct from the body.

### Ubiquitous overexpression of Gnmt increases SAH and decreases SAM levels in larvae
Because increased Gnmt abundance in old eyes correlated with increased SAH levels, we next examined whether Gnmt overexpression was sufficient to increase SAH levels in flies. GNMT expression correlates positively with SAH levels in human hepatic carcinoma cell lines (Li et al., 2019; Wang et al., 2011a) and knockdown of Gnmt in *Drosophila* is sufficient to decrease SAH levels in whole flies (Obata et al., 2014); however, the impact of Gnmt overexpression on SAH levels has not been investigated in flies. To determine whether increased Gnmt is sufficient to increase SAH levels in *Drosophila*, we overexpressed Gnmt in larvae under control of the ubiquitous *tub-Gal4* driver and examined levels of SAM and SAH by targeted metabolite analysis (Fig. 2A). As a control, we examined the $F_1$ progeny of *tub-Gal4* flies crossed with $w^{1118}$ (+). We also overexpressed a mutant form of Gnmt with lower catalytic activity (Gnmt$^{S145A}$) as an additional control (Obata and Miura, 2015). Serine 145 is a conserved residue in mammals and *Drosophila* that is proposed to interact with the carboxyl group of SAM, and a substitution of this residue to an alanine in mammalian Gnmt results in an 85% reduction of specific activity (DebRoy et al., 2013). In *Drosophila*, the S145A mutant has decreased catalytic activity but its overexpression does not negatively impact expression or function of endogenous Gnmt (Obata and Miura, 2015). When we examined Gnmt protein levels in extracts from the Gnmt-overexpressing larvae relative to control, we observed higher levels of both Gnmt and Gnmt$^{S145A}$ relative to endogenous Gnmt (Fig. 2B-D). While both overexpression lines showed significantly increased levels of Gnmt, only the wild-type Gnmt overexpression significantly reduced SAM levels and increased SAH levels (Fig. 2E,F). Thus, increased expression of Gnmt is sufficient to increase SAH levels in *Drosophila* and requires its full catalytic activity.

### Photoreceptor-specific overexpression or knockdown of Gnmt does not impact age-dependent retinal degeneration
Manipulation of one-carbon metabolism or its associated pathways can extend lifespan across multiple species (Lionaki et al., 2022) and fat-body-specific overexpression of Gnmt extends lifespan in flies (Obata and Miura, 2015). Thus, we next investigated whether the increased expression of Gnmt in old eyes protected against age-dependent retinal degeneration. The median lifespan of flies is close to 55 days post-eclosion (D55), and wild-type strains show stochastic retinal degeneration beginning at D50 and becoming more severe by D60 (Escobedo et al., 2023). We overexpressed Gnmt in adult photoreceptors under the control of *Rh1-Gal4* and assessed retinal degeneration by optic neutralization of adult male flies (Fig. 3A). As a control, we examined the $F_1$ progeny of *Rh1-Gal4* flies crossed to $w^{1118}$ (+). We observed a loss of approximately 2% of rhabdomeres by D40 in both the Gnmt overexpression and control conditions with no significant difference in the percentage of missing rhabdomeres between genotypes at any age (Fig. 3B).

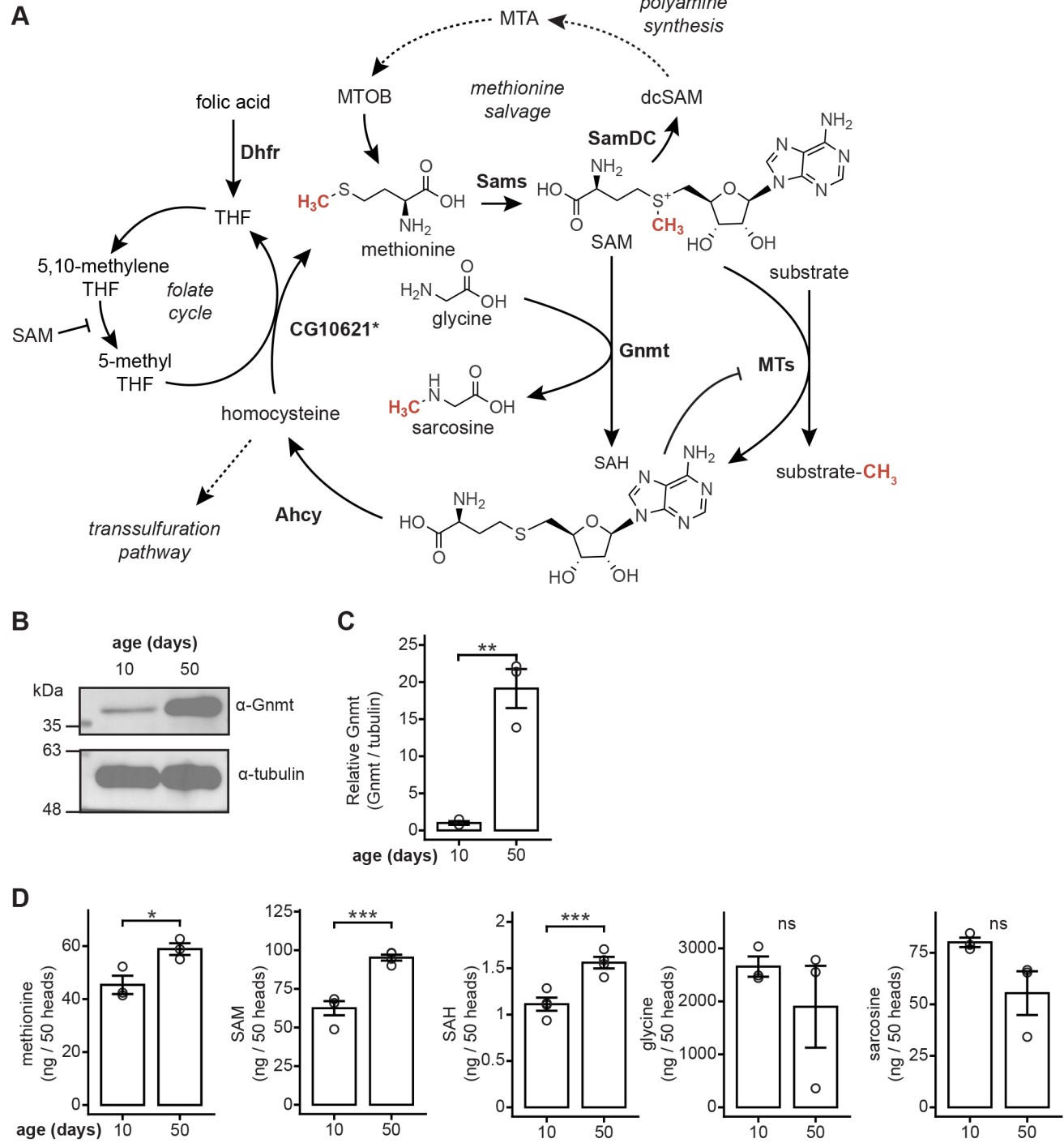

**Fig. 1. Gnmt increase in heads of old *Drosophila* is accompanied by changes in methionine cycle metabolites.** (A) Simplified diagram of *Drosophila* one-carbon metabolism. Asterisk indicates that CG10621 is the homolog of the enzyme predicted to catalyze this reaction. Ahcy, Adenosylhomocysteinase; dcSAM, decarboxylated SAM; Dhfr, Dihydrofolate reductase; Gnmt, Glycine-*N*-methyltransferase; MTA, 5′-methylthioadenosine; MTOB, 4-methylthio-2-oxobutanoic acid; MTs, methyltransferases; SAM, *S*-adenosylmethionine; Sams, *S*-adenosylmethionine synthase; SAH, *S*-adenosylhomocysteine; SamDC, *S*-adenosylmethionine decarboxylase; THF, 5,6,7,8 tetrahydrofolate. (B) Western blot of Gnmt in head extracts from 10- and 50-day-old *OregonR* males. Tubulin is shown as a loading control on the same blot. (C) Bar graph showing quantification of Gnmt abundance as in B. Graph depicts mean±s.d. with individual replicates overlaid as circles ($n=3$). **$P<0.01$ (unpaired two-tailed Student's *t*-test). (D) LC-MS/MS analysis of metabolite levels in head extracts from 10- and 50-day-old *OregonR* males. Graph depicts mean±s.d. with individual replicates overlaid as circles ($n=4$ SAM, SAH; $n=3$ methionine, sarcosine, glycine). ns, not significant ($P>0.05$); *$P<0.05$, ***$P<0.001$ (unpaired two-tailed Student's *t*-test).

Because overexpression did not protect against retinal degeneration, we next examined whether decreased expression of Gnmt would exacerbate age-dependent retinal degeneration. To test this, we knocked down Gnmt in adult photoreceptors by expressing shRNA against Gnmt (*Gnmt* RNAi) under the control of *Rh1-Gal4* (Fig. 3C). As a control, we expressed a non-specific shRNA (*mCherry* RNAi). We confirmed the effectiveness of the RNAi in larvae and observed a 90% decrease in *Gnmt* transcript levels by qPCR and a decrease in

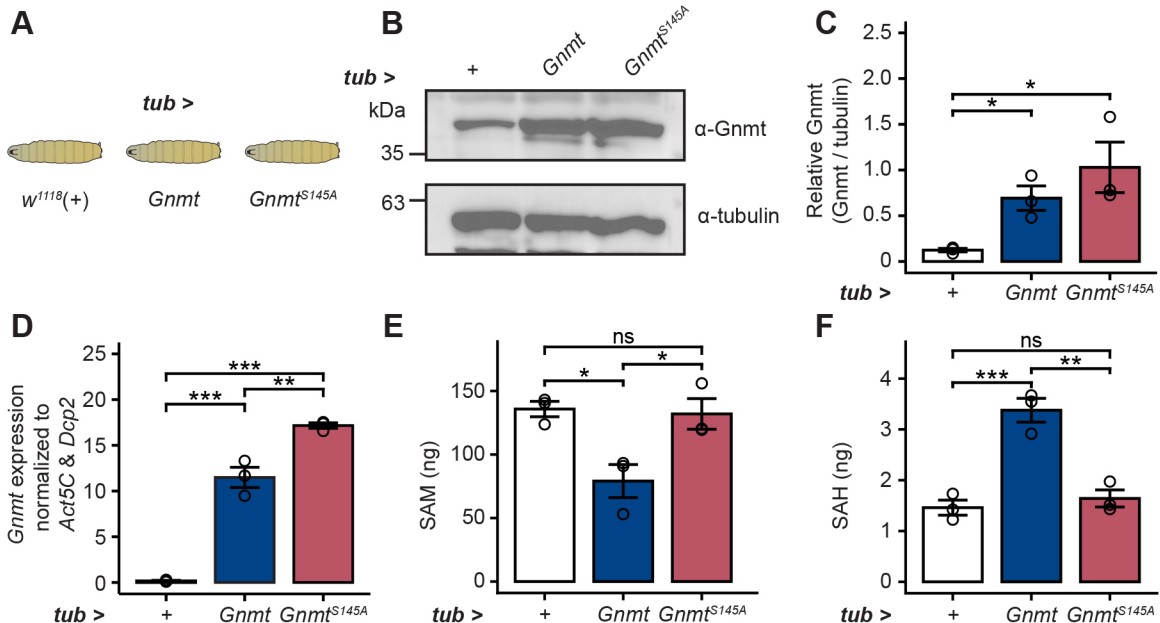

**Fig. 2. Ubiquitous overexpression of Gnmt increases SAH and decreases SAM levels in larvae.** (A) Schematic of Gnmt overexpression in larvae expressing Gnmt or Gnmt$^{S145A}$ under *tub-Gal4* control. Driver outcrossed to $w^{1118}$ (+) was used as control. (B) Western blot of Gnmt in larvae. Tubulin is shown as a loading control on the same blot. (C) Bar graph showing quantification of Gnmt abundance as in B. Graph depicts mean±s.d. with individual replicates overlaid as circles (*n*=3). *$P$<0.05 (one-way ANOVA with Tukey's post-hoc test). (D) qRT-PCR showing expression of Gnmt. Relative mRNA levels (expression) were calculated as the geometric mean of starting quantity normalized to the housekeeping genes *Actin 5C* (*Act5C*) and *Decapping protein 2* (*DCP2*). Graph depicts mean±s.d. with individual replicates overlaid as circles (*n*=3). *$P$<0.05 (one-way ANOVA with Tukey's post-hoc test). (E,F) LC-MS/MS analysis of SAM and SAH levels in larvae. Graph depicts mean±s.d. with individual replicates overlaid as circles (*n*=3). *$P$<0.05, **$P$<0.01, ***$P$<0.001 (one-way ANOVA with Tukey's post-hoc test). ns, not significant.

protein abundance by western blotting analysis compared to the mCherry RNAi control (Fig. S1). We observed a similar loss of 1% of rhabdomeres by D50 in both the *Gnmt* RNAi and the *mCherry* RNAi control with no significant difference in the percentage of missing rhabdomeres between genotypes at any age (Fig. 3D). Thus, we conclude that photoreceptor-autonomous expression of Gnmt does not protect against age-dependent retinal degeneration.

## Overexpression of Gnmt alters rhythmic gene expression in photoreceptors

We next explored whether increased Gnmt expression could have other functional consequences in the aging eye. In old photoreceptors, we previously observed drastic changes in gene expression and chromatin, including global decreases in histone methylation (Hall et al., 2017; Jauregui-Lozano et al., 2023). The gene expression changes in aging photoreceptors include major alterations in rhythmic expression (McGovern et al., 2026). Since another enzyme involved in one-carbon metabolism, Ahcy, is necessary for circadian-regulated gene expression in mouse cells (Greco et al., 2020), we hypothesized that Gnmt regulates rhythmic gene expression in the *Drosophila* eye. To determine whether increased Gnmt alters rhythmic gene expression, we overexpressed Gnmt in the eye under control of the *LongGMR-Gal4* (*lGMR-Gal4*) driver (Escobedo et al., 2021) and profiled gene expression in photoreceptors every 4 h beginning at zeitgeber time (ZT) 0 across a circadian day-night cycle, 12 h of light followed by 12 h of dark (LD conditions), in D10 male flies (Fig. 4A). To profile the photoreceptor transcriptome, we isolated nuclear RNA from photoreceptor cells by immuno-enriching nuclei tagged with GFP$^{KASH}$ expressed under control of the *Rh1* (*ninaE*) promoter as described previously (Jauregui-Lozano et al., 2021). As controls, we profiled the F$_1$ progeny from the *lGMR-Gal4, Rh1-GFP$^{KASH}$*

flies outcrossed to either $w^{1118}$ (+) or *UAS-Gnmt$^{S145A}$* and performed western blotting to confirm an increase in protein abundance (Fig. 4B). Additionally, we measured levels of SAM, SAH, methionine, glycine and sarcosine in heads of these flies but did not observe significant changes in any of these metabolites (Fig. S2A). Despite strong expression of *lGMR-Gal4* in the eye, there was a relatively small increase in Gnmt in the entire head relative to that observed in larvae using *tub-Gal4*. Therefore, we suspect any difference in metabolite levels was minimized due to the dilution of the metabolites by additional head tissue with normal levels of Gnmt. As expected from our previous overexpression experiment, we did not observe any retinal degeneration by D10 in wild-type Gnmt-overexpressing flies or control (Fig. S2B).

We obtained photoreceptor RNA-seq data in triplicate for each ZT to profile changes to the rhythmic transcriptome in all three genotypes. Principal component analysis separated flies by genotype and ZT with the wild-type Gnmt-overexpression flies separating from both controls (Fig. 4C). We identified genes with altered rhythmic expression using dryR, a statistical framework designed to assess differential rhythmicity (Weger et al., 2021). dryR separates genes into rhythmicity categories based on rhythmic parameters, such as phase and amplitude of expression. We identified 7806 genes that met a statistical cutoff for accurate model fitting when comparing three conditions [Bayesian Information Criterion weight (BICW)≥0.5; Table S3]: 4571 of these genes were arrhythmic and 1555 genes exhibited the same rhythmic expression pattern in all three genotypes (Fig. 4D). A further 1441 genes showed alterations in rhythmicity upon overexpression of wild-type Gnmt, Gnmt$^{S145A}$, or both Gnmt and Gnmt$^{S145A}$. Of these, overexpression of wild-type Gnmt, but not Gnmt$^{S145A}$, altered the rhythmicity of 899 genes: 433 genes lost rhythmicity, 344 genes gained rhythmicity and 122 genes in the 'change' category were altered in amplitude or phase relative to

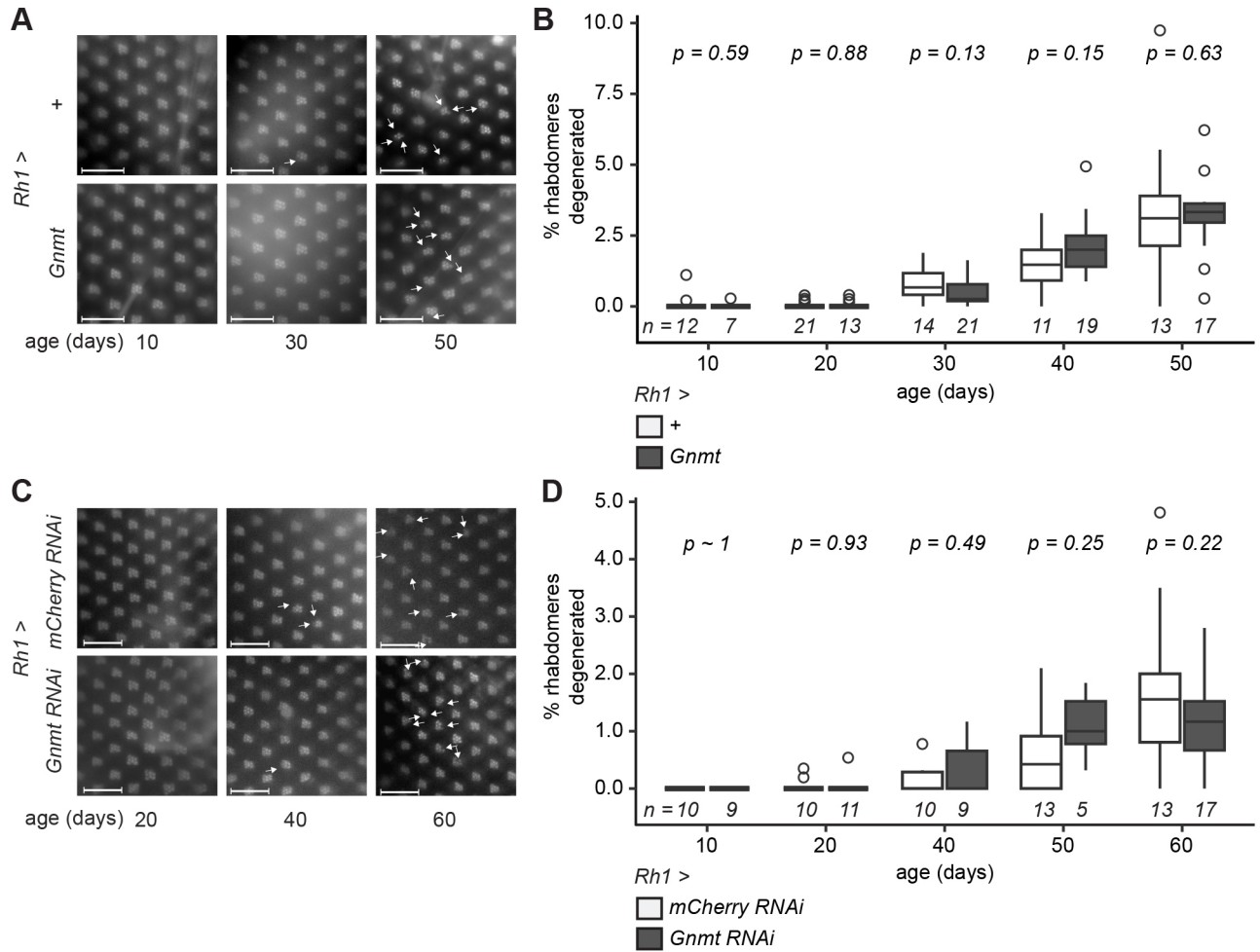

**Fig. 3. Photoreceptor-specific overexpression or knockdown of Gnmt does not impact age-dependent retinal degeneration.** (A,C) Representative images of optic neutralization in male flies at the indicated ages. Gnmt or shRNA specific for Gnmt or mCherry were expressed under *Rh1-Gal4* control. Plus sign denotes driver outcrossed to *w[1118]*. Gnmt RNAi #3 (BDSC #53282; Table S1) was used for *Gnmt* RNAi. White arrows indicate missing rhabdomeres. Scale bars: 25 µm. (B,D) Boxplots showing quantification of data from A and C. Lower and upper hinges correspond to the first and third quartiles, and the whiskers extend to the smallest or largest values no more than 1.5× inter-quartile range (IQR) from each hinge. Horizontal lines represent the median. Outliers are represented as circles. *n* is indicated per condition. *P*-values determined by Welch's *t*-test comparing both conditions at each age.

either control. Genes within this wild-type Gnmt-dependent rhythmicity category were likely sensitive to the increased activity of Gnmt because overexpression of Gnmt[S145A] did not alter their rhythmic expression. However, some genes exhibited overlapping changes when we overexpressed either wild-type Gnmt or Gnmt[S145A]: 84 genes lost, 139 genes gained, and 44 genes were altered in amplitude or phase under both conditions. Because the overexpression of proteins can induce stress, we cannot rule out that changes shared by both forms of Gnmt resulted from a non-specific response to protein overexpression. Intriguingly, some genes exhibited rhythm alterations only upon Gnmt[S145A] expression, including 196 genes that lost and 79 genes that gained rhythmicity. Thus, Gnmt overexpression results in changes in rhythmic gene expression that are largely, but not entirely, associated with its catalytic activity.

The genes with altered patterns of rhythmic expression upon overexpression of Gnmt peaked in rhythmicity during distinct times of day (Fig. 4E). Genes that lost or gained rhythmicity in response to wild-type Gnmt overexpression typically peaked during the early light (ZT0-ZT4) and early dark (ZT12-ZT16) phases. In contrast, genes that lost rhythmicity in Gnmt[S145A] flies peaked

in late light (ZT8-ZT12) and late dark (ZT20-ZT0) while genes that gained rhythmicity peaked only in the early light (ZT0-ZT4) phase. Gene ontology term analysis showed that wild-type Gnmt overexpression affected the rhythmicity of genes associated with the electron transport chain, Wnt signaling, H3K9 tri-methylation, behavior and long-term memory (Fig. 4F). Interestingly, both forms of Gnmt affected the rhythmicity of genes associated with amino sugar metabolic processes and calcium ion transmembrane transport. Overexpression of Gnmt[S145A], but not wild-type Gnmt, affected the rhythmicity of genes associated with nucleoside phosphate and cellular lipid metabolism as well as protein folding. Rhythmic gene expression is controlled in part by the activity of the molecular clock (Patke et al., 2020). We therefore examined whether changes in rhythmic gene expression could be caused by alterations in the rhythmic expression of core clock genes. Surprisingly, we did not observe differences in the rhythms of members of the molecular clock TTFL, such as Clock (*Clk*), period (*per*), timeless (*tim*), PAR-domain protein 1 (*Pdp1*), cryptochrome (*cry*) and vrille (*vri*) (Fig. 4G). All molecular clock genes were still categorized as rhythmic in wild-type Gnmt overexpressing flies and control. We also note that neither Gnmt

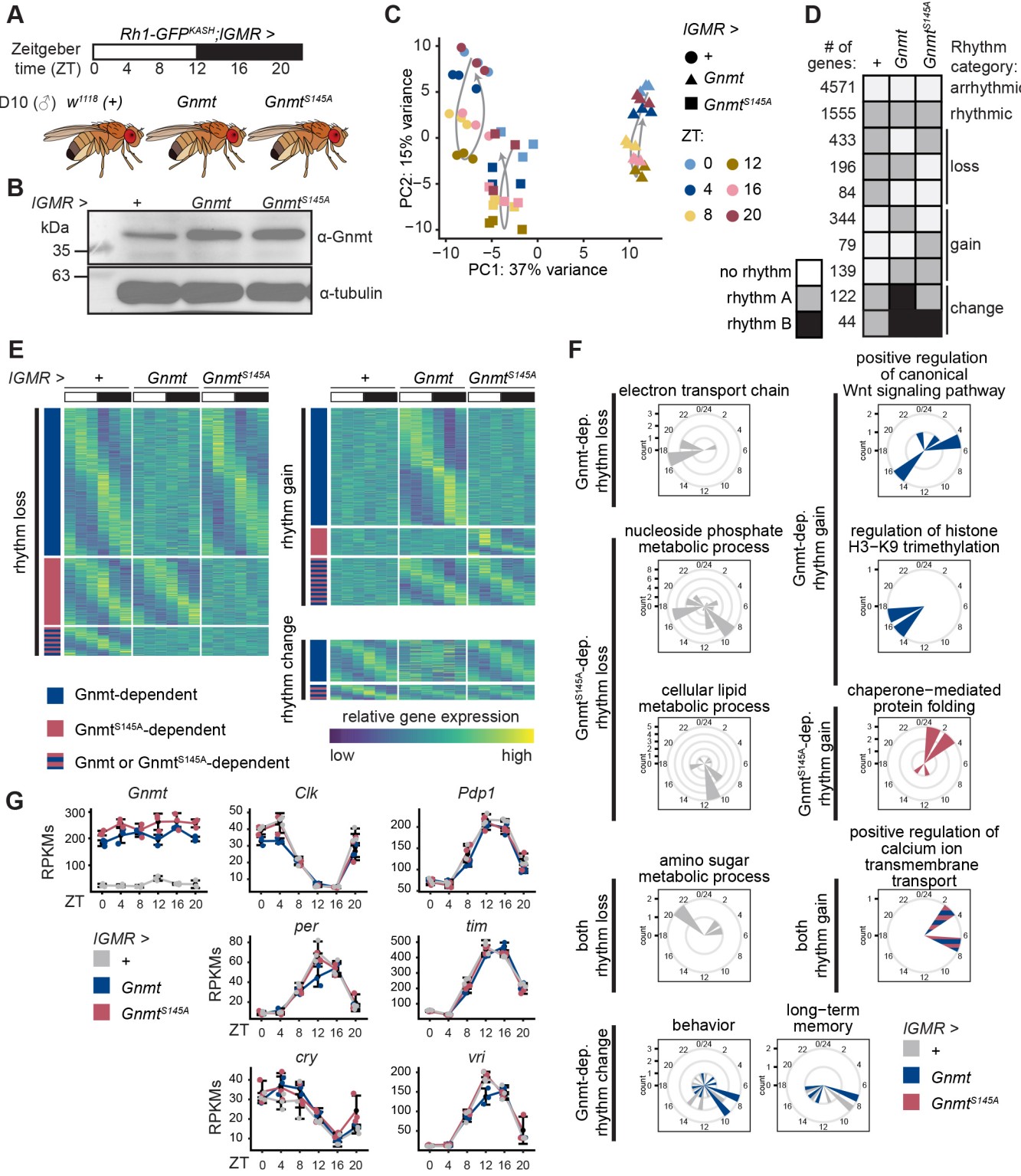

**Fig. 4. Overexpression of Gnmt alters rhythmic gene expression in photoreceptors.** (A) Diurnal photoreceptor nuclear RNA-seq analysis in D10 *Rh1-GFP^KASH* male flies overexpressing Gnmt or Gnmt^S145A under *LongGMR-Gal4* (*IGMR*) control (*n*=3). Driver outcrossed to *w^1118* (+) was used as control. Samples were collected at the indicated time of day relative to the onset of the light cycle (zeitgeber time, ZT). (B) Western blot of Gnmt in D10 male heads. Tubulin is shown as a loading control on the same blot. (C) Principal component analysis of RNA-seq samples based on variance-stabilized gene expression levels. (D) Heatmap showing rhythmic dryR categories (BICW≥0.5) with at least 40 genes. Color indicates whether genes are not rhythmic (white), rhythmic (gray) or have an altered rhythm (black). Genes are broadly categorized into models that lose, gain or have change in rhythmicity upon overexpression of Gnmt or Gnmt^S145A. (E) Heat maps depicting *z*-score of relative expression. (F) Radial histograms depicting ZTs of peak expression for genes of select enriched gene ontology terms in each rhythmicity category. dep., dependent. (G) Line plots of selected molecular circadian clock gene expression and *Gnmt*. Individual RPKM data points shown with error bars representing the mean±s.d. All depicted genes are classified as rhythmic under all conditions.

nor Gnmt$^{S145A}$ expression under *lGMR-Gal4* control exhibited rhythmic expression (Fig. 4G), suggesting that the observed changes to rhythmic gene expression were due to increased steady state levels of Gnmt rather than a periodic transcriptional or translational stress induced by overexpression of a protein. Thus, Gnmt overexpression disrupts rhythmic gene expression under LD conditions but does not alter the phase or amplitude of genes associated with the circadian clock. Our data indicate that the catalytic activity of Gnmt influences rhythmic gene expression under the circadian day-night cycle but that this role may not be dependent on transcription of the molecular clock.

### Overexpression of Gnmt alters overall gene expression across the day

In addition to rhythmicity, Gnmt has been implicated more generally in the regulation of gene expression. For instance, expression of Gnmt inhibits the expression of multiple oncogenes in HepG2 cells (Simile et al., 2022) and Gnmt interacts with the promoters of transcription factors in mouse livers to modulate their expression (Chang et al., 2018). Therefore, we assessed whether Gnmt could also impact overall gene expression levels in the eye. To address this, we examined differential gene expression when either form of Gnmt was overexpressed relative to control in averaged gene expression across all time points (ZT0-ZT20) using DESeq2 (Love et al., 2014) (Tables S4 and S5). Overexpression of either form of Gnmt resulted in similar numbers of differentially expressed genes: wild-type Gnmt overexpression upregulated 268 and downregulated 192 genes (Fig. 5A) and Gnmt$^{S145A}$ overexpression upregulated 194 and downregulated 165 genes (Fig. 5B). Of these, 257 genes were significantly differentially expressed only upon wild-type Gnmt overexpression, indicating that these genes are susceptible to increased Gnmt activity (Fig. 5C). However, 189 genes were significantly differentially expressed when either form of Gnmt was overexpressed. Much like the rhythmic gene expression analysis, genes that respond similarly to either form of Gnmt may be indicative of a non-specific response to protein overexpression. This is supported by many of the shared upregulated genes being related to stress response, such as *wntD* and the heat shock proteins *Hsp70Ba* and *Hsp70Bb*. Further, 156 genes were differentially expressed only upon Gnmt$^{S145A}$ expression. These data indicate that increased Gnmt catalytic activity alters both rhythmic gene expression as well as gene expression throughout the day.

Gene ontology term analysis of the differential expression categories showed that wild-type Gnmt altered the expression of genes associated with 'cellular response to heat' as well as 'defense response to Gram-positive bacterium' (Fig. 5D). Overexpression of either form of Gnmt resulted in the downregulation of genes associated with the detection of chemical stimulus. In contrast, Gnmt$^{S145A}$ affected genes associated with 'protein refolding', 'polytene chromosome puffing', 'response to heat' and 'chitin-based cuticle development'.

To test whether the rhythmicity changes observed upon Gnmt overexpression correlated with the overall gene expression changes, we compared these differentially expressed genes with the rhythmicity categories. We observed little overlap between the genes that were differentially expressed and those that had altered rhythmicity (Fig. 5E,F). For instance, only 15 genes that lost rhythmicity in response to wild-type Gnmt overexpression were downregulated, while only 15 genes that gained rhythmicity were upregulated. Thus, Gnmt overexpression has distinct effects on overall gene expression and rhythmic gene expression.

### Eye-specific overexpression Gnmt decreases repressive but not active histone methylation in photoreceptors

Expression level of Gnmt correlates negatively with methyltransferase activity, including DNA methylation (Wang et al., 2011a) and tRNA methylation (Kerr and Heady, 1974), and knockout of Gnmt increases H3K27 trimethylation (me3) levels in mice (Martínez-Chantar et al., 2008). Methyltransferases, including histone lysine methyltransferases, play important roles in gene expression (Hyun et al., 2017), and several histone methyltransferases are involved in regulation of rhythmic gene expression, such as the H3K4 methyltransferase MLL1 (Katada and Sassone-Corsi, 2010). Thus, we next investigated whether Gnmt overexpression decreased histone methylation levels in the eye. To test whether increased Gnmt decreases histone methylation in photoreceptors, we overexpressed Gnmt in the entire eye under the control of *lGMR-Gal4* driver and profiled multiple histone methyl marks using cleavage under target and release using nuclease (CUT&RUN) on immuno-enriched photoreceptor nuclei (Fig. 6A). As a control, we outcrossed flies to $w^{1118}$ (+). We examined H3K4me3 and H3K4 monomethylation (me1) as histone methyl marks associated with active transcription at promoters and enhancers, respectively. Additionally, we examined H3K9me3 and H3K27me3 as histone methyl marks associated with constitutive and facultative heterochromatin, respectively. We examined at least three biological replicates and compared the average levels of each mark in both conditions (Fig. 6B). We identified 4831 H3K4me3, 1593 H3K4me1, 636 H3K9me3 and 1926 H3K27me3 peaks in our control samples (+), and then compared average levels of these peaks between the control and Gnmt-overexpressing flies (Fig. 6C, Table S6). Average H3K4me3 levels were not noticeably altered by Gnmt overexpression, while H3K4me1 levels were only slightly lower relative to the control at the called peaks. However, average levels of both H3K9me3 and H3K27me3 were lower in the Gnmt-overexpressing flies relative to control (Fig. 6C). This decrease in H3K9me3 and H3K27me3 levels in the Gnmt-overexpressing flies was also apparent when we examined individual gene regions in either condition and was observed at the majority of peaks (Fig. 6C). Thus, overexpression of Gnmt decreases levels of repressive but not active histone methyl marks, indicating that some methylation events are more responsive to increased Gnmt levels than others in these cells.

Next, we examined whether the presence of these histone methyl marks correlated with the observed changes in rhythmicity. We first defined the percentage of genes in each rhythmicity category for which promoters or gene bodies overlapped with the peaks called for each histone mark (Fig. 6D). Relatively few expressed genes contained peaks for H3K9me3 or H3K27me3, consistent with the role of these marks in repression. In contrast, many genes were marked by H3K4me3 or H3K4me1. For H3K4me1, H3K9me3 and H3K27me3, the percentage of genes with these marks remained consistent, regardless of rhythmicity category. H3K4me3 peaks were more variable; more genes in the Gnmt-dependent loss, gain and change categories were marked with H3K4me3 compared to those that remained rhythmic. Thus, the decreases in H3K9me3 and H3K27me3 upon Gnmt overexpression do not correlate with rhythmicity changes at a gene-specific level. Because global levels of H3K4me3 and H3K4me did not change, we considered that there may be local changes in genes that had altered rhythmicity in response to Gnmt overexpression. However, when we visualized average levels of these marks across genes in these categories, the patterns and levels of H3K4me3 and H3K4me1 remained consistent in both genotypes in all categories (Fig. S3). Additionally, H3K9me3 and H3K27me3 levels decreased regardless of

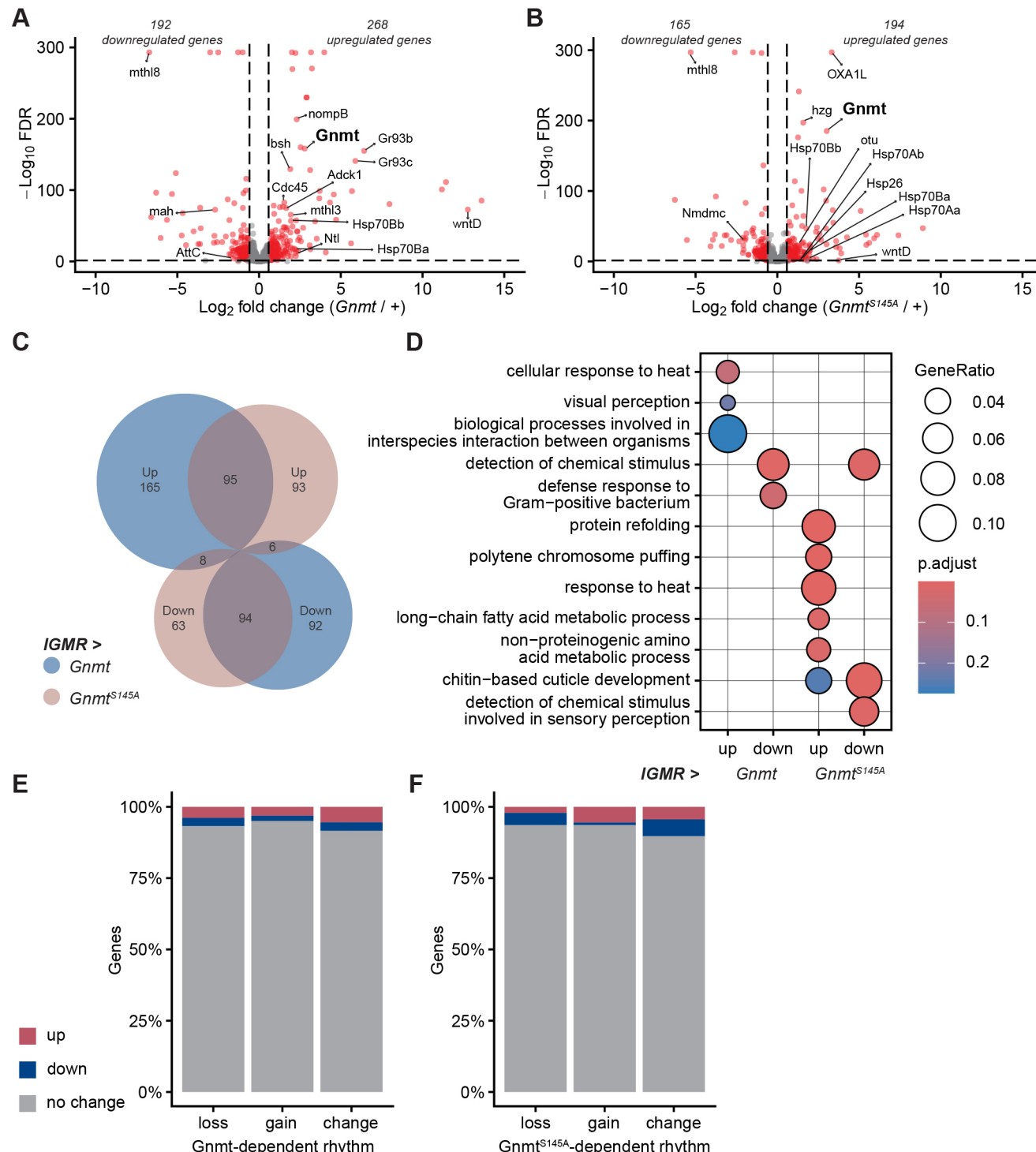

**Fig. 5. Overexpression of Gnmt alters overall gene expression across the day.** (A,B) Volcano plots depicting differential gene expression between *IGMR>Gnmt* and *IGMR>+* or *IGMR>Gnmt^{S145A}* and *IGMR>+* across all ZTs. Differential gene expression was determined using DESeq2 (*n*=18). Individual genes are plotted as points with selected genes labeled. Red points denote genes significant change in expression (FDR<0.05 and |fold change|>1.5). (C) Venn diagram comparing the number of differentially expressed genes when Gnmt or Gnmt^{S145A} are overexpressed. (D) Gene ontology term analysis of genes that are upregulated or downregulated when Gnmt or Gnmt^{S145A} are overexpressed. (E,F) Bar plots showing the relative percentage of genes in rhythmicity categories that are significantly upregulated or downregulated when Gnmt or Gnmt^{S145A} are overexpressed.

category. Similarly, we did not observe any correlation between the genes that were differentially expressed across all time points and those that contained peaks for H3K9me3 or H3K27me3 (Fig. 6E). However, the percentage of differentially expressed genes marked by H3K4me3, and to a lesser extent H3K4me1, were lower than

genes that did not change when Gnmt was overexpressed. Thus, genes marked with H3K4me3 were less likely to be differentially expressed in response to increased Gnmt, indicating that genes within chromatin environments defined by low H3K4me3 may be more susceptible to Gnmt-dependent changes in gene expression.

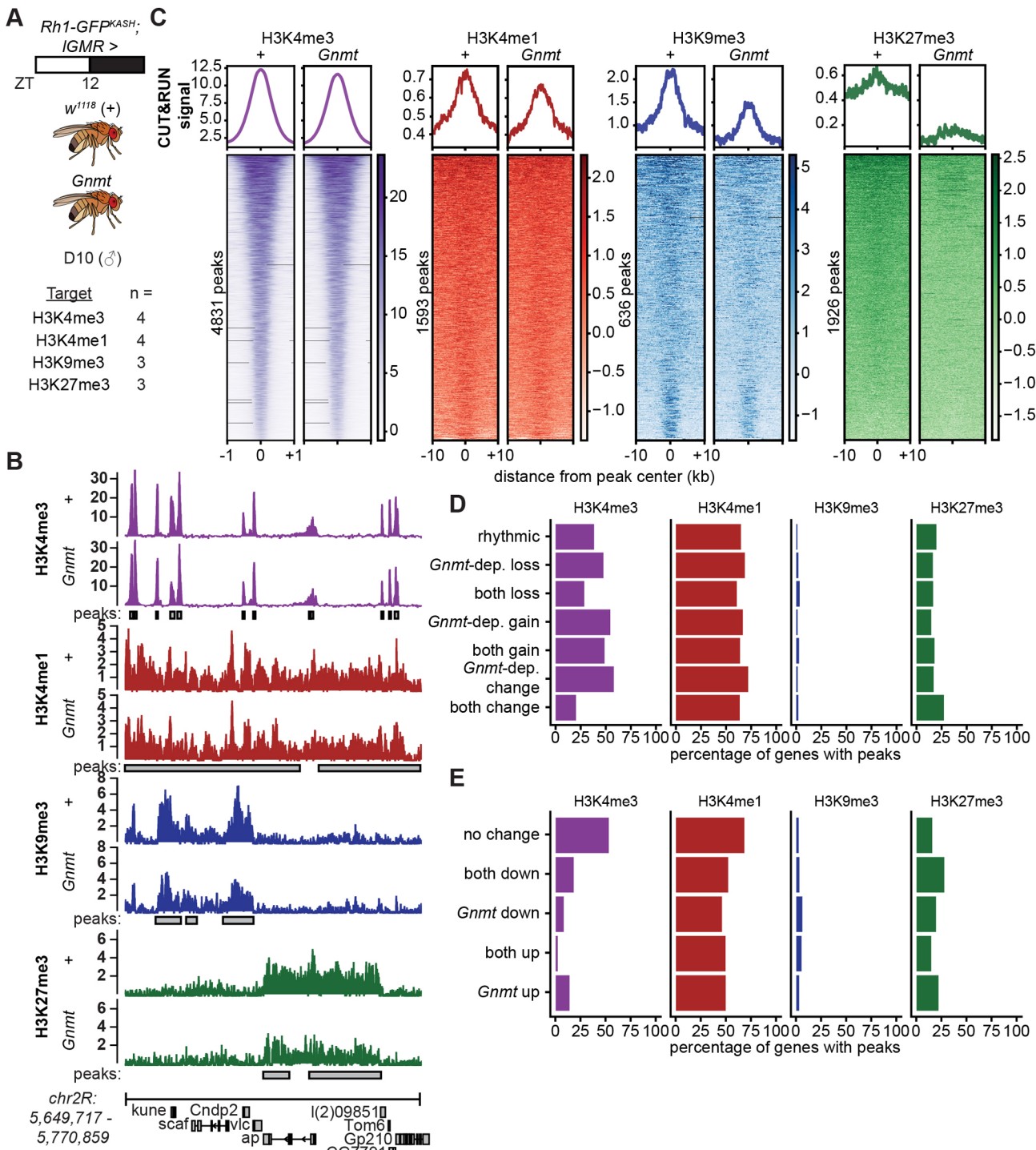

**Fig. 6. Eye-specific overexpression Gnmt decreases repressive but not active histone methylation in photoreceptors.** (A) CUT&RUN analysis of histone methyl marks was performed in photoreceptor nuclei isolated from D10 *Rh1-GFP^KASH* male flies expressing Gnmt under *IGMR* control at ZT12. Driver outcrossed to *w^1118* (+) was used as control. (B) IgG-normalized CUT&RUN tracks at a select region on chromosome 2R. Identified peaks for each mark are shown as horizontal gray bars. Data are mean (H3K4me3 and H3K4me1 *n*=4; H3K9me3 and H3K27me3 *n*=3). Heatmaps are colored by relative signal (*z* score) across the same region (rows) between control (+) and Gnmt overexpression. (C) Gene metaplots and heatmaps showing IgG-normalized average CUT&RUN signal at peaks called for the indicated histone methyl marks (*n*=3). (D,E) Bar plots depicting the proportion of genes in each expression or rhythmicity category for which promoters contain peaks for each mark (promoters and gene bodies).

## Gnmt overexpression in clock cells does not alter circadian locomotor behavior

Since overexpression of Gnmt altered rhythmic gene expression in photoreceptors but did not impair expression of the core clock genes, we next examined whether Gnmt overexpression would alter

circadian behavior. Photoreceptors are an abundant cell type in the fly head that express all members of the molecular clock and play roles in photoentrainment (Schlichting, 2020). *tim* is one of the molecular clock genes responsible for light-dependent regulation of rhythmic gene expression (Kaneko et al., 2000). Other molecular

clock-expressing cells include pacemaker neurons in the brain that are required for maintaining circadian rhythms in the absence of light. Thus, to elucidate the impact of Gnmt on circadian behavior, we overexpressed Gnmt in all *tim*-expressing cells under the control of *tim-Gal4* and measured locomotor activity of individual flies. To determine whether the catalytic activity of Gnmt impacted circadian behavior, we also assayed flies that overexpressed Gnmt$^{S145A}$, as well as *tim-Gal4*, *UAS-Gnmt* and *UAS-Gnmt$^{S145A}$* flies outcrossed to $w^{1118}$. We also assayed flies that expressed unrelated enzymes, β-galactosidase (*UAS-lacZ*) and firefly luciferase (*UAS-pLuc*) with respective controls, to determine whether any changes to locomotor behavior were specific to Gnmt. We measured activity over 5 days of LD conditions followed by 4 days in total darkness (DD) (Fig. 7A, Fig. S4A). The DD measurements allowed us to determine whether the flies could maintain a consistent circadian period in the absence of an environmental cue – an indication of a functioning circadian clock. The timing and strength of circadian periods were not significantly different during DD conditions in any genotype (Fig. 7B,C, Fig. S4B,C). This is consistent with our observation that Gnmt overexpression did not impact the expression of core clock genes in photoreceptors (Fig. 4G).

We considered that, although overexpressed Gnmt does not impact circadian locomotor behavior, it may have other effects on behavior. Flies exhibit characteristic differences in their activity between day and night that are partially, but not wholly, dependent on the molecular circadian clock (Grima et al., 2004; Sehgal et al., 1994). Elevated night-time activity is associated with hyperexcitation of clock neurons (Sheeba et al., 2008) and *Drosophila* models of neurodegenerative diseases, including Alzheimer's disease (Buhl et al., 2019). While *tim-Gal4* driven Gnmt or Gnmt$^{S145A}$ expression did alter day/night activity differences, the observed changes were also observed in the *lacZ* and *pLuc* controls (Fig. 7D-F, Fig. S4D-F). Since *tim-Gal4* is expressed rhythmically, this approach would result in rhythmic expression of the overexpressed proteins across the day, peaking around ZT14 in most *Drosophila* tissues (Jauregui-Lozano et al., 2023; McGovern et al., 2026; Sehgal et al., 1995). Therefore, the observed phenotypes are likely due to disruption of protein homeostasis in clock cells through the rhythmic expression of transgenic proteins rather than through the catalytic activity of these enzymes. Together, our findings highlight the consequences of increased Gnmt in *Drosophila* neurons, suggesting that the alterations in one-carbon metabolism disrupt rhythmic gene expression without disrupting the circadian clock.

## DISCUSSION
Overall, our data suggest that age-related increases in Gnmt expression have distinct consequences that depend on the tissue. Although increased Gnmt in the fat body is protective (Obata and Miura, 2015), we show that increased expression of Gnmt in photoreceptor neurons is not protective during aging and disrupts gene expression, repressive histone methylation, and light-dependent behavioral rhythms. Some of these differences likely reflect the different roles of one carbon metabolism in the liver, or *Drosophila* fat body, relative to neuron-rich organs such as the eye or brain. While the fat body primarily utilizes one-carbon metabolism to support other synthetic pathways, in eyes one-carbon metabolism has roles in supporting their primary function of light sensing. Owing to the high demand of energy by photoreceptors, oxidative phosphorylation and aerobic glycolysis result in the production of reactive oxygen species in the eye, especially under stress conditions like blue light exposure (Stanhope et al., 2023). Catabolism of SAH produces homocysteine, which is used in the synthesis of the antioxidant glutathione. More recently, Ahcy has been

identified as a redox sensor that protects against light-stress in the eye (Stanhope et al., 2025), further supporting the role of one-carbon metabolism in the eye. This helps contextualize our findings regarding Gnmt in the eye. While not protective in photoreceptors, Gnmt may be upregulated to accommodate the observed age-dependent changes to one-carbon metabolites in the eye or head. Consequently, this increased Gnmt may interfere with other processes. Our data suggest that Gnmt interferes with the rhythmic expression of light-dependent genes independently of the molecular circadian clock.

While the activity of the molecular clock TTFL regulates many processes, it is not responsible for the expression of all rhythmically expressed genes under LD conditions. For example, Schneider 2 cells, which do not express multiple components of the TTFL, still maintain rhythmic oscillations of their transcriptome, proteome and metabolomes with temperature as a zeitgeber (Rey et al., 2018). Importantly, a subset of light-dependent genes exhibits sustained rhythmic expression in response to diurnal light cycles (Wijnen et al., 2006). These light-dependent genes retained rhythmicity in *tim$^{01}$* flies but did not exhibit rhythmicity in flies mutant for no receptor potential A (*norpA*), which lack a functioning phototransduction cascade (Pak et al., 1970). Several genes that lose rhythmicity in the *norpA* mutants also exhibit altered rhythmicity when Gnmt is overexpressed, including *Pka-C3*, *CG31038* and *Inos*. Therefore, our data show that increased Gnmt expression alters photoreceptor rhythmic gene expression without disrupting core clock gene expression, potentially in response to light sensing.

Unlike other one-carbon enzymes, Gnmt does not alter rhythmic gene expression through altered H3K4 methylation, as both location and relative levels of H3K4 methylation were unaffected by Gnmt overexpression. In contrast, repressive histone methylation was noticeably decreased by Gnmt overexpression. This may involve the role of histone proteins as a major methyl sink in eukaryotic cells in response to disrupted methionine metabolism (Ye and Tu, 2018). It is proposed that histones serve as a methyl sink to maintain SAM homeostasis when large SAM consuming processes are inhibited, such as phosphatidylethanolamine (PE) methylation (Ye et al., 2017). Conversely, increasing SAM consumption through the overexpression of PE methyltransferase in mammalian cells reduced levels of multiple histone methyl marks (Ye et al., 2017). In addition to PE methyltransferase, overexpression of other SAM-consuming enzymes with relatively high $K_i$ for SAH, such as nicotinamide-*N*-methyltransferase, reduced histone methylation levels (Eckert et al., 2019; Ulanovskaya et al., 2013). Therefore, increased activity of Gnmt may increase SAM consumption and SAH production, resulting in reduced histone methylation to maintain SAM homeostasis. It remains unclear why H3K4 methylation levels were less susceptible to changes in Gnmt expression. One explanation is a hierarchical order of histone sites as preferential methyl sinks (H3K36>H3K79>H3K4) (Ye et al., 2017). This may explain why H3K4me3 levels were affected by neither Gnmt overexpression nor knockdown of Ahcy in the eye (Stanhope et al., 2025). Further, this hierarchy could be driven by differences in SAM affinity and sensitivity to SAH inhibition of methyltransferases that vary depending on the enzyme and substrate (Richon et al., 2011). Other methylation targets such as tRNA, mRNA or other proteins could also be susceptible to changes in Gnmt expression. In addition to the production of SAH and sarcosine, the catalytic activity of Gnmt likely promotes altered flux of one-carbon metabolites. Increased flux of one-carbon units through both the methionine cycle and methionine salvage has been observed in whole flies at 4 weeks of age (Parkhitko et al., 2021). Consistent with this, we previously observed a significant increase in levels of 5′-methylthioadenosine,

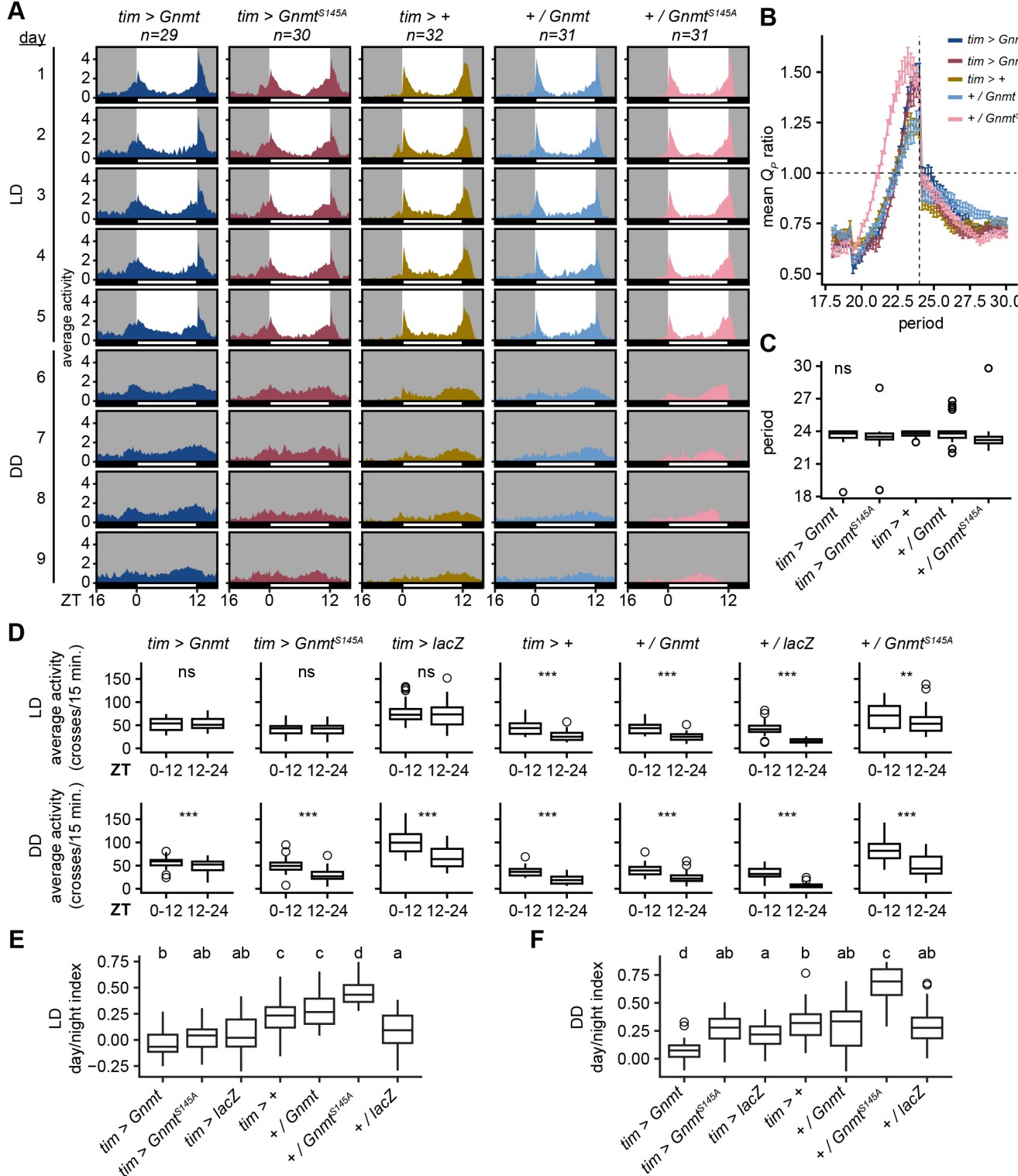

**Fig. 7. Gnmt overexpression in clock cells does not impact circadian locomotor behavior.** (A) Actograms showing averaged activity in flies of the indicated genotypes during light:dark (LD) and dark:dark (DD) conditions. Background color denotes status of the light (white=on, gray=off). Horizontal bar denotes subjective day (white) and night (black) based on ZT. Plus sign indicates lines crossed with $w^{1118}$. $n$ indicated above each plot. (B) Chi-square circadian periodogram analysis of rhythmic locomotor activity under DD conditions. The ratio of observed periodogram power to the significance threshold is plotted as a function of the tested rhythmic period: ($Q_P$ ratio=$Q_P$ activity/$Q_P$ significance). Period testing range=18-30 h with a testing resolution of 0.2 h. Error bars depict mean±s.e.m. Mean $Q_P$ ratio >1 denotes a significant circadian period. Vertical dashed line corresponds with a period of 24 h. (C) Boxplots depicting circadian periods of individual flies under DD conditions. ns, not significant ($P$>0.05; one-way ANOVA). (D) Boxplots depicting average activity (crosses/15 min) of individual flies in either LD or DD conditions. Data are binned into either 'Day' (ZT0-ZT12) or 'Night' (ZT12-ZT24). ns, not significant ($P$>0.05); **$P$<0.01, ***$P$<0.001 (Wilcoxon signed-rank test). *lacZ* encodes β-galactosidase. (E,F) Boxplots depicting day/night index of individual flies in either LD or DD conditions. Different letters indicate statistical significance $P$<0.05 (one-way ANOVA with Tukey's multiple comparisons post-hoc test).

the substrate of the rate-limiting step of methionine salvage (Avila et al., 2004), in 40-day-old eyes (Hall et al., 2021). Increased flux would also explain why we observed increased methionine and SAM levels in aging heads despite an increase in Gnmt activity. It is possible that the subcellular localization of Gnmt could also impact its effect on methylation reactions. Gnmt is found abundantly in *Drosophila* fat body nuclei (Kashio and Miura, 2025). Interestingly, Gnmt is rapidly degraded in the nucleus in response to low SAM availability (Kashio and Miura, 2025). This raises the possibility that age-dependent changes to SAM abundance and other one-carbon metabolites influence levels of nuclear Gnmt in old fly heads and eyes, thereby specifically regulating methylated targets in the nucleus.

Although we had originally assumed that Gnmt catalytic activity would be required to alter methylation potential and thereby regulate gene expression, our data indicate that increased Gnmt has additional consequences that do not involve its increased catalytic activity. While some of the alterations in gene expression that occur in both the wild-type and catalytic mutant overexpression flies may simply represent a nonspecific response to protein overexpression, we also observe distinct gene expression changes in only the catalytic mutant, suggesting that overexpressing Gnmt could have consequences that are not explained by its catalytic role alone. Non-enzymatic roles of Gnmt have been documented in mammals, such as its ability to bind to the folate-derived 5-methylTHF. In the mouse liver, increased Gnmt expression improves intracellular levels and retention of folate by preventing its consumption for use in homocysteine remethylation (Wang et al., 2011b). Folate availability can impact processes such as mRNA m⁶A methylation, an important modification for RNA splicing and translation, in *Drosophila* neurons (Liu et al., 2025). Further, 5-methylTHF is a substrate for methionine synthesis; therefore, Gnmt protein may still influence availability of one-carbon metabolites or flux irrespective of its catalytic activity. In addition to folate binding, Gnmt has been found to associate with chromatin (Chang et al., 2018; Simile et al., 2022). Future investigation of these non-enzymatic functions of Gnmt in *Drosophila* as well as its subcellular-specific roles will help elucidate the mechanisms connecting Gnmt to gene expression and its role in neurons.

## MATERIALS AND METHODS
### *Drosophila* strains and husbandry
*Drosophila* were maintained on standard cornmeal/yeast media at 25°C with 70% humidity under LD conditions where ZT0 indicates the beginning of the light phase. Male flies were collected at the indicated ages (days post-eclosion, D) ±1 day and within ±15 min of ZT12 (or the indicated ZT). Dark-phase collections (ZT12-ZT20) were performed under red light. *UAS-Gnmt* and *UAS-Gnmt^{S145A}* lines were graciously provided to us by Dr Masayuki Miura (Obata et al., 2014). Fly lines used in this study are described in Table S1.

### Protein extraction and western blotting
Protein extracts from five third instar larvae or 50 adult heads were prepared by homogenization in the following buffers: for larvae, 50 mM Tris-HCl (pH 8.0), 150 mM NaCl, 1% NP-40; for heads, 4% SDS, 1 mM EDTA, 75 mM Tris-HCl (pH 6.8). Homogenization buffers were supplemented with 10 mM DTT, 1 mM PMSF, 2 μM pepstatin and 2 μM leupeptin, and 20 μg (larvae) or 25 μg (heads) of total protein was subjected to SDS-polyacrylamide gel electrophoresis and western blotting analysis using the following antibodies: anti-Gnmt (1:1000; provided by Dr Masayuki Miura, National Institute for Basic Biology, Okazaki, Japan) and anti-α-tubulin (1:15,000; DSHB, AA4.3; RRID:AB_579793).

### Targeted metabolite analysis
Metabolite extracts from five third instar larvae or 50 adult heads were analyzed per biological replicate. SAM and SAH were analyzed in the same sample; methionine, sarcosine and glycine were analyzed together. Briefly,

samples were suspended in methanol (SAM and SAH) or PBS (methionine, glycine, sarcosine) and the following internal standards were added: $d_3$-SAM (Toronto Research Chemicals, A291532; 100 ng), $d_4$-SAH (Cayman Chemical, 9000372; 100 ng), $d_3$-methionine (Cayman Chemical, 34826; 100 ng), glycine-$^{13}C^{15}N$ (Toronto Research Chemicals, G615997; 2500 ng) or $d_3$-sarcosine (Toronto Research Chemicals, S140504; 50 ng). Tissues were manually homogenized with a Pellet Mixer (VWR, 47747-370) for 1 min, centrifuged at 4°C for 10 min at 16,000 *g* and the supernatant was collected for LC-MS/MS analysis. For SAM and SAH, samples were analyzed as described previously (Stanhope et al., 2025). For methionine, glycine and sarcosine, an Agilent 1290 Infinity II liquid chromatography (LC) system coupled to an Agilent 6470 series QQQ mass spectrometer (MS/MS) was used to analyze samples. (Agilent Technologies). An Imtakt Intrada Amino Acid 2.0 mm×150 mm, 3 μm column was used for LC separation (Imtakt USA). The buffers were: (A) acetonitrile+0.3% formic acid and (B) 100 mM ammonium formate:acetonitrile (80:20 v/v). The linear LC gradient was as follows: time 0 min, 20% B; time 5 min, 20% B; time 11 min, 35% B; time 20 min, 100% B; time 22 min, 100% B; time 22.5 min, 20% B; time 30 min, 20% B. The flow rate was 0.3 ml min⁻¹. Multiple reaction monitoring was used for MS analysis according to Table S7. Data were acquired in positive electrospray ionization (ESI) mode. The jet stream ESI interface had a gas temperature of 325°C, gas flow rate of 9 l min⁻¹, nebulizer pressure of 35 psi, sheath gas temperature of 250°C, sheath gas flow rate of 7 l min⁻¹, capillary voltage of 3500 V in positive mode, and nozzle voltage of 1000 V. The ΔEMV voltage was 400 V. Agilent Masshunter Quantitative analysis software was used for data analysis (version 10.1).

### Optic neutralization
Retinal degeneration was assessed in individual live flies by bright-field microscopy using optic neutralization (Franceschini and Kirschfeld, 1971). Images were de-identified and scored for missing rhabdomeres with the assessor unaware of groupings.

### Quantitative RT-qPCR
RNA was extracted from five larvae using the Zymoprep Direct-zol RNA MicroPrep kit (Zymo Research, R2050). cDNA was generated from 665 ng of RNA using Episcript Reverse Transcriptase (Epicentre) and random hexamer primers. Primers are listed in Table S2.

### Nuclei immunoenrichment-based RNA-seq and CUT&RUN
Photoreceptor nuclei were immuno-enriched from the heads of progeny from *Rh1-GFP^{KASH};lGMR-Gal4* crossed to the indicated lines using GFP antibodies coupled to magnetic beads as previously described (Jauregui-Lozano et al., 2021). Detailed protocols for nuclei immunoenrichment (NIE)-based RNA-Seq and CUT&RUN are available (doi:10.17504/protocols.io.kxygxp1d4l8j/v3; Jauregui-Lozano et al., 2021).

### RNA-seq
Libraries were generated from nuclear RNA using the Tecan Ovation SoLo RNA-seq library preparation kit with *D. melanogaster* AnyDeplete probes to eliminate rRNA, and sequencing was performed on the Illumina HiSeq 2000 platform.

### CUT&RUN
CUT&RUN was performed on bead-bound nuclei as described by Meers et al. (2019a) using pAG-MNase fusion protein as described by McGovern et al. (2026). The following antibodies were used for CUT&RUN with 20 min of pAG-MNase activation time: anti-IgG (EpiCypher, 13-0042; RRID:AB_2923178), anti-H3K4me3 (EpiCypher, 13-0060), anti-H3K4me1 (Abcam, ab8895; RRID:AB_306847), anti-H3K9me3 (Abcam, ab176916; RRID:AB_2797591), anti-H3K27me3 (EpiCypher, 13-0055; RRID:AB_3665059). CUT&RUN libraries were prepared using the NEBNext® Ultra™ II DNA Library Prep Kit for Illumina® with modifications to enrich for small fragments. Libraries were sequenced on the Illumina NovaSeq X Plus platform.

### RNA-seq and CUT&RUN data analysis
We obtained at least 28 million reads for RNA-seq samples and 6 million reads for CUT&RUN samples. All samples were normalized to library size

during analysis [i.e. counts per million (CPM) and reads per kilobase million (RPKM)].

## Alignment and initial data processing

Data were trimmed using Trimmomatic (Bolger et al., 2014) for paired-end reads, and aligned to the *D. melanogaster* genome assembly (BDGP6.46 – release 113) using HISAT2 (Kim et al., 2019) for RNA-seq data or bowtie2 (Langmead and Salzberg, 2012) for CUT&RUN data with the –*dovetail* argument. For RNA-seq data, the first 5 bp from the forward read was removed prior to trimming using the argument HEADCROP:5, as recommended for Ovation SoLo libraries. Sequence Alignment/Map (SAM) files were processed using SAMtools (Danecek et al., 2021) to generate sorted, deduplicated Binary Alignment/Map (BAM) files. CPM-normalized bigWig files were generated using deepTools (Ramírez et al., 2016).

## RNA-seq

Subread featureCounts (Liao et al., 2019) was used to count exonic features. Photoreceptor-expressed genes were defined as CPM$\geq$1 in all three replicates for a single ZT or in one genotype. We identified differentially expressed genes across all time points (ZT) using DESeq2 (Love et al., 2014). RPKMs were determined using edgeR (Chen et al., 2025). Differential rhythmic gene expression analysis was performed using dryR (Weger et al., 2021) on exonic reads with a cutoff of BICW$\geq$0.5. Phase and amplitude values were determined using dryR.

## CUT&RUN

Averaged, IgG-subtracted bigWig files were created by subtracting IgG negative controls from their respective biological replicates and averaging the resulting bigWigs using deepTools. H3K4me3 peaks were called on deduplicated BAM files with IgG controls using MACS3 at an FDR cutoff of 0.0001 (Zhang et al., 2008). H3K4me1, H3K9me3, H3K27me3 peaks were called on BAM files with IgG controls using SEACR in the relaxed mode (Meers et al., 2019b). We selected peaks that were present in at least two biological replicates of the control samples (*lGMR>+*) using the BEDTools multiinter function, merging H3K4me peaks within 1000 bp and H3K27me3 peaks within 3000 bp. H3K9me3 and H3K4me3 peaks were not merged. Only peaks mapped to chromosomes 2L, 2R, 3L, 2R, 4, X and Y were retained.

## Data visualization, custom GTF, and gene ontology

Gene ontology analysis for differentially expressed genes was performed using clusterProfiler (Yu et al., 2012). Custom plots were generated in R (version 4.4.0). Venn diagrams were generated using BioVenn (Hulsen et al., 2008). CUT&RUN data were CPM-normalized and subtracted for respective IgG control for visualization. CUT&RUN data were visualized across genomic coordinates using Gviz (Hahne and Ivanek, 2016). Heatmaps and metaplots were generated using deepTools based on custom GTF files representing unique used transcripts with blacklisted regions filtered. Custom GTFs and the custom CUT&RUN blacklist were created using the *D. melanogaster* genome assembly (BDGP6.46 – release 113) as described previously (McGovern et al., 2026). Gene ontology analysis for rhythmicity categories was performed using GOrilla (Eden et al., 2007, 2009).

## Behavioral assays

Circadian behavioral assays were performed as described by Buhl et al. (2019). Briefly, 32 male D5 flies were placed in 65 mm glass tubes with agar/sucrose food. Locomotor activity was monitored with DAM2 *Drosophila* Activity Monitors (TriKinetics Inc.) in an incubator maintained at 25°C with light control. Activity was measured in 15-min bins over the course of 5 days under 12:12 LD conditions followed by 4 days under DD. Data were analyzed using the ShinyR-DAM package (Cichewicz and Hirsh, 2018) and the MATLAB Phase, Activity, and Sleep under Entrainment (PHASE) package (Persons et al., 2022). Individuals that died over the course of the assay were excluded from analysis. The day/night index was calculated as (day average activity−night average activity)/(day average activity+night average activity) to compare the difference in day versus night activity between conditions as described by Kumar et al. (2012).

## Acknowledgements

We thank Amber Jannasch and the Bindley Bioscience Center Metabolite Profiling Facility for assistance with targeted metabolite analysis. We thank Dr Masayuki Miura for providing the UAS-Gnmt stocks and anti-Gnmt antibody used in this study. Stocks obtained from the Bloomington *Drosophila* Stock Center (NIH P40OD018537) were used in this study. CUT&RUN sequencing analysis was carried out in the Center for Medical Genomics at Indiana University School of Medicine, which is partially supported by the Indiana University Grand Challenges Precision Health Initiative. The authors gratefully acknowledge the support of from the Institute for Cancer Research, NIH grant P30 CA023168

## Competing interests

The authors declare no competing or financial interests.

## Author contributions

Conceptualization: S.D.L., V.M.W.; Data curation: S.D.L.; Formal analysis: S.D.L., M.N.M., D.M.K.; Funding acquisition: V.M.W.; Project administration: V.M.W.; Supervision: V.M.W.; Visualization: S.D.L.; Writing – original draft: S.D.L., M.N.M.; Writing – review & editing: V.M.W.

## Funding

This work was supported by the National Institutes of Health [R01EY033734 and R21AG094190 to V.M.W.]. The content is solely the responsibility of the authors and does not necessarily represent the official views of the National Institutes of Health. Additional support was provided by Bird Stair funding from the Purdue University Department of Biochemistry to S.D.L. Open Access funding provided by Purdue University. Deposited in PMC for immediate release.

## Data and resource availability

High-throughput sequencing data are available at the Gene Expression Omnibus (GEO) under accession numbers: GSE267056 (RNA-seq; 54 samples); GSE305323 (CUT&RUN; 59 samples). Custom codes used for generating the plots or analysis in this study are available upon request. All other relevant data and details of resources can be found within the article and its supplementary information.

## First Person

This article has an associated First Person interview with the first author of the paper.

## Peer review history

The peer review history is available online at https://journals.biologists.com/jcs/lookup/doi/10.1242/jcs.264529.reviewer-comments.pdf

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
