## [Peer Review File · Journal of Cell Science]

Increased Glycine-N-methyltransferase expression disrupts light-dependent gene expression rhythms in the *Drosophila* eye

Seth D. Lammert, Makayla N. Marlin, Danny M. Kanj and Vikki M. Weake

DOI: 10.1242/jcs.264529

Editor: Caroline Hill

Review timeline

Original submission:	27 October 2025
Editorial decision:	23 December 2025
First revision received:	5 March 2026
Accepted:	15 March 2026

Original submission

First decision letter

MS ID#: jcs.264529

MS TITLE: Increased Glycine-N-methyltransferase expression disrupts light-dependent rhythms in the *Drosophila* eye.

AUTHORS: Seth D. Lammert; Makayla N. Marlin; Danny M. Kanj; Vikki M Weake

ARTICLE TYPE: Research Article

Dear Dr Weake,

We have now reached a decision on the above manuscript.

As you will see, the reviewers gave favourable reports but raised some critical points that will require amendments to your manuscript. I hope that you will be able to carry these out because I would like to be able to accept your paper, depending on further comments from reviewers.

Reviewer 1

Advance summary and potential significance to field

In this manuscript, the authors investigate the effects of overexpressing Glycine-N-methyltransferase (Gnmt) in the photoreceptors of the *Drosophila* eye disc during the third instar larval stage and later developmental stages. They also examine a catalytically inactive mutant form of Gnmt. To characterize the consequences of this overexpression, the authors employ several genome-wide approaches, including RNA-seq and CUT&RUN, along with behavioral assays to assess circadian rhythms.

The study reports that Gnmt overexpression alters rhythmic gene expression without affecting core clock gene expression or photoreceptor viability. They further show that Gnmt overexpression reduces global repressive histone methylation, while largely sparing H3K4 methylation. Based on these findings, the authors propose that Gnmt selectively inhibits particular methylation reactions and disrupts rhythmic gene expression independently of H3K4 methylation. Finally, they

demonstrate that overexpression of Gnmt in clock cells suppresses day-night differences in locomotor activity under diurnal light conditions.

In my opinion, the results presented are informative, and the experiments are well executed and appropriate.

Comments for the author

Major comments [Please request additional experiments only if they are essential for supporting the conclusions; authors should be encouraged to highlight any claims that are preliminary or speculative, or to discuss any pitfalls or alternative interpretations in a 'Limitations' section]

Minor comments

A main concern arises from the observation that many of the phenotypes produced by overexpressing wild-type Gnmt are also observed when the catalytically inactive mutant is overexpressed. This suggests, as the author mentioned, that at least some of the reported effects may be partially independent of Gnmt's enzymatic activity. As a result, it becomes difficult to distinguish which phenotypes arise from Gnmt's metabolic role in one-carbon metabolism and which reflect catalytic-independent functions of the protein. This ambiguity complicates the overall interpretation of the findings, since the metabolic effects cannot be clearly separated from other potential structural or regulatory consequences of Gnmt overexpression.

Related to this point, the manuscript raises the question of how general the observed disruptions in rhythmic gene expression truly are. Are these effects specific to Gnmt, or might similar changes be detected when overexpressing other proteins unrelated to one-carbon metabolism or circadian regulation? In other words, does the mutant condition used by the authors reveal something unique about Gnmt's function, or could overexpression of an unrelated protein produce comparable alterations simply due to perturbation of photoreceptor homeostasis? Including such controls—or at least addressing this possibility—would strengthen the authors' conclusions regarding the specificity of Gnmt's role in regulating rhythmic gene expression.

Minor problem.

I believe that Figure 6 is incorrectly labeled: panel C should be labeled as B, and panel B should be labeled as C.

Reviewer 2

Advance summary and potential significance to field

In this study "Increased Glycine-N-methyltransferase expression disrupts light-dependent rhythms the *Drosophila* eye," the authors investigated the consequences of Glycine-N-methyltransferase (Gnmt) overexpression in circadian rhythm. Gnmt regulates levels of the methionine pathway metabolites SAM and SAH and additionally impacts the intersection between the folate cycle and one-carbon metabolism. To advance the field, the authors of this work investigated the consequences of overexpression of wild-type or a catalytically inactive versions of Gnmt in different *Drosophila* tissues with a focus on expression in eye photoreceptors. The authors determine that while either overexpression or RNAi knockdown in photoreceptors did not impact age related retinal degeneration, gene expressions patterns including those subject to circadian rhythms were disrupted by overexpression of both the wild-type and catalytically inactive versions of Gnmt. The flies with overexpression also had altered circadian behaviors. Interesting, the authors found that histone methylation patterns were differently affected, with little change to histone H3H4 methylation while repressive histone marks H3K9 and H3K27 levels were reduced relative to control flies. Stated conclusions are supported by the data. The results will be of particular interest to those working to understand the role of Gnmt in the control of circadian rhythm behaviors

Comments for the author

Major comments [Please request additional experiments only if they are essential for supporting the conclusions; authors should be encouraged to highlight any claims that are preliminary or speculative, or to discuss any pitfalls or alternative interpretations in a 'Limitations' section]

Minor comments

1. In the legend for Figure 4, Line 806-807, the duplicate text "Radial histograms depict ZTs of peak gene expression" in the part for E should be removed.
2. In the legend for Figure 5, I believe that the letter labels are incorrect. The bar plots are E, F rather than C,D; the Venn diagram C rather than E and the GO term analysis C rather than F.
3. In the legend for Figure 6, I believe that that the letter labels are incorrect. The gene metaplots are C rather than B and the CUT&RUN tracks are B rather than C.
4. On page 14, line 340-341, the sentence "Under LD conditions...other controls (Fig. 7E), is duplicated and one should be removed.
5. As altered expression of Gnmt impacts SAM and SAH levels, it would be valuable if the authors could include information about histones as a reservoir for methylation (Ye and Tu, 2018 as example) and how this possibility could influence patterns of methylation & gene expression.

First revision

Author response to reviewers' comments

We would like to thank the editor and the reviewers for their comments and suggestions. We have addressed each comment with our specific responses below (*italics*). Changes to the text are indicated on the tracked-changes version with **yellow highlights**, and indicated by the line numbers in our responses below. We have included parts of the relevant text in quotation marks in the reviewer response to make it easier to identify those sections.

As a brief overview, we have replaced main figures 4 and 7 with new versions. We have also fixed minor issues with the other figures, so have uploaded replacement versions of all the main figures. We have added a new supplemental figure 4. We have made minor edits to the text, particularly to the abstract, summary statement, the figure legends, results, and discussion to address the concerns of both reviewers. Figure color schemes have been adjusted to fit colorblind-friendly standards.

Reviewer #1

In this manuscript, the authors investigate the effects of overexpressing Glycine-N-methyltransferase (Gnmt) in the photoreceptors of the *Drosophila* eye disc during the third instar larval stage and later developmental stages. They also examine a catalytically inactive mutant form of Gnmt. To characterize the consequences of this overexpression, the authors employ several genome-wide approaches, including RNA-seq and CUT&RUN, along with behavioral assays to assess circadian rhythms. The study reports that Gnmt overexpression alters rhythmic gene expression without affecting core clock gene expression or photoreceptor viability. They further show that Gnmt overexpression reduces global repressive histone methylation, while largely sparing H3K4 methylation. Based on these findings, the authors propose that Gnmt selectively inhibits particular methylation reactions and disrupts rhythmic gene expression independently of H3K4 methylation. Finally, they demonstrate that overexpression of Gnmt in clock cells suppresses day-night differences in locomotor activity under diurnal light conditions. In my opinion, the results presented are informative, and the experiments are well executed and appropriate.

Comment 1:

A main concern arises from the observation that many of the phenotypes produced by overexpressing wild-type Gnmt are also observed when the catalytically inactive mutant is overexpressed. This suggests, as the author mentioned, that at least some of the reported effects may be partially independent of Gnmt's enzymatic activity. As a result, it becomes

difficult to distinguish which phenotypes arise from Gnmt's metabolic role in one-carbon metabolism and which reflect catalytic-independent functions of the protein. This ambiguity complicates the overall interpretation of the findings, since the metabolic effects cannot be clearly separated from other potential structural or regulatory consequences of Gnmt overexpression.

Related to this point, the manuscript raises the question of how general the observed disruptions in rhythmic gene expression truly are. Are these effects specific to Gnmt, or might similar changes be detected when overexpressing other proteins unrelated to one-carbon metabolism or circadian regulation? In other words, does the mutant condition used by the authors reveal something unique about Gnmt's function, or could overexpression of an unrelated protein produce comparable alterations simply due to perturbation of photoreceptor homeostasis? Including such controls—or at least addressing this possibility—would strengthen the authors' conclusions regarding the specificity of Gnmt's role in regulating rhythmic gene expression.

We thank the reviewer for raising this important concern, and we have made several changes to the text to discuss this point both in our rhythmic RNA-seq analysis (Figure 4) and added in several additional controls to our circadian behavior analysis in Figure 7. We have adjusted our interpretation of the data and conclusions accordingly throughout the text to make it clear where our results include controls that enable us to attribute dependence on Gnmt catalytic activity, and where our results may potentially be due to non-specific protein overexpression.

First, our experimental design for the rhythmic RNA-seq analysis does enable us to identify those genes that require Gnmt catalytic activity by comparing the impact of overexpressing wild-type Gnmt versus the catalytic mutant S145A. Indeed, overexpression of WT Gnmt but not Gnmt[S145A] alters rhythmicity of 899 genes. However, we agree with this reviewer that those genes that show similar changes in rhythmicity upon overexpression of WT and mutant Gnmt (84 genes lose, 139 genes gain, 44 genes change) are potentially responding simply to non-specific protein overexpression. We remain a bit puzzled by the genes that show alterations in rhythmic expression only upon overexpression of the S145A mutant Gnmt, but not WT Gnmt (196 genes lose; 79 genes gain). These alterations seem unlikely due to non-specific protein overexpression because if so, they should also have been observed in the WT Gnmt overexpressing flies. Because Gnmt is known to bind folate-derived 5-methylTHF, it is possible that overexpression of a catalytically inactive mutant Gnmt alters flux through one-carbon metabolism in a manner that would differ from the impact of overexpressing WT Gnmt. However, we have not examined this possibility in this study. We have chosen to describe those genes that respond only to the S145A mutant in Figure 4 because of the potential for Gnmt to impact flux independent of its catalytic activity, and we include some text in the discussion to address this point. However, we now focus more strongly on the aspects of Gnmt function that can be clearly assigned to its catalytic activity based on our analysis. Relevant sections of the text have been edited throughout the manuscript, but the two sections quoted below summarize the major changes.

Line 201: “Genes within this wild-type Gnmt-dependent rhythmicity category were likely sensitive to the increased activity of Gnmt because overexpression of GnmtS145A did not alter their rhythmic expression. However, some genes exhibited overlapping changes when we overexpressed either wild-type Gnmt or GnmtS145A; 84 genes lost, 139 genes gained, and 44 genes were altered in amplitude or phase under both conditions. Because the overexpression of proteins can induce stress, we cannot rule out that changes shared by both forms of Gnmt resulted from a non-specific response to protein overexpression.”

Line 415 (discussion): “While some of the alterations in gene expression that occur in both the wild-type and catalytic mutant overexpression flies may simply represent a nonspecific response to protein overexpression, we also observe distinct gene expression changes in only the catalytic mutant, suggesting that overexpressing Gnmt could have consequences that are not explained by its catalytic role alone.”

Second, upon reflection, we realized that our circadian behavioral analysis lacked controls to determine if the day/night changes in activity (which are reasonably subtle in any case) were

specific to *Gnmt*, or might as this reviewer suggests, result from overexpression of any protein. The potential for non-specific rhythmic impacts on behavior assays is relatively high because in the circadian behavioral assays shown in Figure 7, we used the *tim-Gal4* driver - which is expressed in a rhythmic pattern throughout the day. We note that in our RNA-seq analysis, we used the eye-specific driver *longGMR-Gal4*, which does not drive rhythmic *Gnmt* expression (Fig. 7G). To address this point, we added in two additional controls to these circadian behavior experiments consisting of UAS-transgenes expressing the following enzymes: *E. coli* β -galactosidase (*lacZ*) or *P. pyralis* luciferase (*pLuc*). Similar to *Gnmt* overexpression, these transgenes do not impact circadian behavior strength or period. However, just like WT or *S145A* mutant *Gnmt*, overexpression of either *lacZ* or *pLuc* suppresses the differences in day/night activity. Thus, we now conclude that the behavioral phenotypes we observed upon *Gnmt* overexpression were likely a non-specific consequence of protein (potentially enzyme?) overexpression. We now include these new data in Figure 7 panels D, E, and F, and show the *Luc* data in a new supplemental figure, Figure S4. These changes are outlined in the text (see lines 325 - 350). We would like to really thank the reviewer for suggesting this control since this turned out to be extremely important for the overall conclusions of our study. We think it remains important to keep these circadian behavior data in the manuscript, even though they are largely negative, because there are several studies that have reported day/night differences in circadian behavior using this exact genetic approach. We plan to revisit this analysis for other genes and drivers, and test if this problem is limited to enzymes overexpressed under the rhythmic *tim-Gal4* driver, or would be caused by neuronal-specific expression of any transgene (for example, under *elav-Gal4* or *nSyb-Gal4*).

We outline some of the specific changes to the manuscript below:

- a) We removed our previous statement regarding *Gnmt*'s impact on day/night activity from the abstract because similar phenotypes were observed in our new controls (see new abstract).
- b) We removed previous emphasis of *Gnmt*'s activity phenotype in the summary statement (see new summary statement).
- c) We adjusted the concluding statement in our introduction (Line 98-100) to better reflect the findings in context of the additional behavioral controls:

"However, our data show that overexpression of Gnmt in clock cells did not disrupt circadian locomotor behavior, suggesting that increased expression of Gnmt in photoreceptors impacts light-dependent rhythmic gene expression but not the molecular clock."

- d) We have changed the title of Figure 7 title to "*Gnmt* overexpression in clock cells does not alter circadian locomotor behavior".
- e) We included the following statement at Line 324 to describe the additional controls for our behavioral assays:

"To determine if the catalytic activity of Gnmt impacted circadian behavior we also assayed flies that overexpressed GnmtS145A, as well as tim-Gal4, UAS-Gnmt, and UAS-GnmtS145A flies outcrossed to w1118. We also assayed flies that expressed unrelated enzymes, β -galactosidase (UAS-lacZ) and firefly luciferase (UAS-pLuc) with respective controls, to determine if any changes to locomotor behavior were specific to Gnmt"

- f) In our original manuscript, we focused more on the day/night activity phenotypes. However, given the results of the controls we reduced the scope of focus on this section. To facilitate this, we added a transition statement at line 335:

"We considered that while overexpressed Gnmt does not impact circadian locomotor behavior, it may have other effects on behavior."

- g) And we also replaced a majority of the end of Figure 7's text at line Line 342 with the

following text to better reflect that these data are negative:

*“Since *tim-Gal4* is expressed rhythmically, this approach would result in rhythmic expression of the overexpressed proteins across the day, peaking around ZT14 in most *Drosophila* tissues (Jauregui-Lozano et al., 2023; McGovern et al., 2026; Sehgal et al., 1995). Therefore, the observed phenotypes are likely due to disruption of protein homeostasis in clock cells through the rhythmic expression of transgenic proteins rather than through the catalytic activity of these enzymes. Together, our findings highlight the consequences of increased *Gnmt* in *Drosophila* neurons, suggesting that the alterations in one-carbon metabolism disrupt rhythmic gene expression without disrupting the circadian clock.”*

*h) In Figure 7, we added data for *tim > lacZ* and its control to panels D,E,F to show that the observed activity phenotypes are not unique to *Gnmt*. To reduce redundancy and simplify the figures, we also created an additional supplemental Figure S4, that includes data from the new behavioral controls: actograms of *tim > lacZ*, *tim > Luc*, + / *LacZ*, + / *pLuc* (Panel A). Periodogram (Panel B). Boxplots of calculated periods (Panel C). LD and DD average activity boxplots of *tim > pLuc*, + > *pLuc* (Panel D). LD and DD Day night indexes comparing *pLuc* to *Gnmt* expression (Panels E,F).*

Comment #3: I believe that Figure 6 is incorrectly labeled: panel C should be labeled as B, and panel B should be labeled as C.

We have corrected the panel labels/figure legend for Figure 6.

Reviewer 2:

In this study "Increased Glycine-N-methyltransferase expression disrupts light-dependent rhythms the *Drosophila* eye," the authors investigated the consequences of Glycine-N- methyltransferase (*Gnmt*) overexpression in circadian rhythm. *Gnmt* regulates levels of the methionine pathway metabolites SAM and SAH and additionally impacts the intersection between the folate cycle and one-carbon metabolism. To advance the field, the authors of this work investigated the consequences of overexpression of wild-type or a catalytically inactive versions of *Gnmt* in different *Drosophila* tissues with a focus on expression in eye photoreceptors. The authors determine that while either overexpression or RNAi knockdown in photoreceptors did not impact age related retinal degeneration, gene expressions patterns including those subject to circadian rhythms were disrupted by overexpression of both the wild- type and catalytically inactive versions of *Gnmt*. The flies with overexpression also had altered circadian behaviors. Interesting, the authors found that histone methylation patterns were differently affected, with little change to histone H3H4 methylation while repressive histone marks H3K9 and H3K27 levels were reduced relative to control flies. Stated conclusions are supported by the data. The results will be of particular interest to those working to understand the role of *Gnmt* in the control of circadian rhythm behaviors

Comment #1: In the legend for Figure 4, Line 806-807, the duplicate text "Radial histograms depict ZTs of peak gene expression" in the part for E should be removed.

We have corrected the figure legend for Figure 4 (line 843).

Comment #2: In the legend for Figure 5, I believe that the letter labels are incorrect. The bar plots are E, F rather than C,D; the Venn diagram C rather than E and the GO term analysis C rather than F.

We have corrected the figure legends for Figure 5 (line 849).

Comment #3: In the legend for Figure 6, I believe that that the letter labels are incorrect. The gene metaplots are C rather than B and the CUT&RUN tracks are B rather than C.

We have corrected the panel labels/figure legend for Figure 6.

Comment #4: On page 14, line 340-341, the sentence "Under LD conditions...other controls (Fig.

7E), is duplicated and one should be removed.

We have removed the duplicated sentence.

Comment #5: As altered expression of Gnmt impacts SAM and SAH levels, it would be valuable if the authors could include information about histones as a reservoir for methylation (Ye and Tu, 2018 as example) and how this possibility could influence patterns of methylation & gene expression.

We thank the reviewer for bringing our attention to this study, and we agree this is a very important point of discussion. We have added a paragraph to the discussion (see below) regarding this study and how it may relate to our observations.

Discussion (line 384): “This may involve the role of histone proteins as a major methyl sink in eukaryotic cells in response to disrupted methionine metabolism (Ye and Tu, 2018). It is proposed that histones serve as a methyl sink to maintain SAM homeostasis when large SAM consuming processes are inhibited, such as phosphatidylethanolamine (PE) methylation (Ye et al., 2017). Conversely, increasing SAM consumption through the overexpression of PE methyltransferase in mammalian cells reduced levels of multiple histone methyl marks (Ye et al., 2017). In addition to PE methyltransferase, overexpression of other SAM-consuming enzymes with relatively high K_i 's for SAH, such as nicotinamide-N-methyltransferase, reduced histone methylation levels (Eckert et al., 2019; Ulanovskaya et al., 2013). Therefore, increased activity of Gnmt may increase SAM consumption and SAH production, resulting in reduced histone methylation to maintain SAM homeostasis. It remains unclear why H3K4 methylation levels were less susceptible to changes in Gnmt expression. One explanation is a hierarchical order of histone sites as preferential methyl sinks (H3K36 > H3K79 > H3K4) (Ye et al., 2017). This may explain why H3K4me3 levels were neither affected by Gnmt overexpression nor knockdown of Ahcy in the eye (Stanhope et al., 2025). Further, this hierarchy could be driven by differences in SAM affinity and sensitivity to SAH inhibition of methyltransferases that vary depending on the enzyme and substrate (Richon et al., 2011).”

Second decision letter

MS ID#: jcs.264529R1

MS Title: Increased Glycine-N-methyltransferase expression disrupts light-dependent gene expression rhythms in the Drosophila eye.

Authors: Seth D. Lammert; Makayla N. Marlin; Danny M. Kanj; Vikki M Weake

Article Type: Research Article

Dear Dr Weake,

I am happy to tell you that your manuscript has been accepted for publication in Journal of Cell Science, pending standard publication integrity checks.